# Acute Changes in Body Muscle Mass and Fat Depletion in Hospitalized Young Trauma Patients: A Descriptive Retrospective Study

**DOI:** 10.3390/diseases11030120

**Published:** 2023-09-11

**Authors:** Hassan Al-Thani, Bianca M. Wahlen, Ayman El-Menyar, Mohammad Asim, Lena Ribhi Nassar, Mohamed Nadeem Ahmed, Syed Nabir, Monira Mollazehi, Husham Abdelrahman

**Affiliations:** 1Department of Surgery, Trauma & Vascular Surgery, Hamad Medical Corporation, Doha P.O. Box 3050, Qatar; althanih@hotmail.com; 2Department of Anesthesiology, Hamad Medical Corporation, Doha P.O. Box 3050, Qatar; bwahlen@googlemail.com; 3Clinical Research, Trauma & Vascular Surgery Section, Hamad Medical Corporation, Doha P.O. Box 3050, Qatar; masim1@hamad.qa; 4Department of Clinical Medicine, Weill Cornell Medicine, Doha P.O. Box 24144, Qatar; 5Department of Dietetics and Nutrition, Hamad Medical Corporation, Doha P.O. Box 3050, Qatar; lnassar@hamad.qa; 6Department of Radiology, Hamad Medical Corporation, Doha P.O. Box 3050, Qatar; drnadeem76@gmail.com (M.N.A.); snabir@hotmail.com (S.N.); 7Trauma Registry, Trauma Surgery, Hamad Medical Corporation, Doha P.O. Box 3050, Qatar; mollazehi@hamad.qa; 8Department of Surgery, Trauma Surgery, Hamad Medical Corporation, Doha P.O. Box 3050, Qatar; hushamco@hotmail.com

**Keywords:** sarcopenia, muscle loss, fat depletion, trauma, CT scan, abdominal injury, nutrition

## Abstract

**Background**: Loss of muscle mass, and its strength, is associated with adverse outcomes in many medical and surgical conditions. Trauma patients may get malnourished during their hospital course due to many interrelated contributing factors. However, there is insufficient knowledge on the acute muscle and fat changes in young trauma patients in the early days post-admission. Objective: to explore the diagnosis, feeding status, and outcome of muscle mass loss among young abdominal polytrauma patients. **Methods**: It was a retrospective study including hospitalized abdominal trauma patients who underwent an abdominal computerized tomographic (CT) examination initially and a follow-up one week later. CT scan-based automatic and manual analysis of the muscles and fat of the abdominal region was calculated and compared. Also, we evaluated the feeding and nutritional values to explore the adequacy of the provided calories and proteins and the potential influence of enteral feeding on the CT-based parameters for muscle loss and fat depletion. **Results**: There were 138 eligible subjects with a mean age of 32.8 ± 13.5 years; of them, 92% were males. Operative interventions were performed on two-thirds of the patients, including abdominal surgery (43%), orthopedic surgeries (34%), and neurosurgical procedures (8.1%). On admission, 56% received oral feeding, and this rate slightly increased to 58.4% after the first week. Enteral feed was prescribed for the remaining, except for two patients. The percentage of change in the total psoas muscle area was significantly reduced after one week of admission in patients on enteral feed as compared to those in the oral feeding group (*p* = 0.001). There were no statistically significant differences in the percentages of changes in the CT scan findings except for the total psoas muscle area (*p* = 0.001) and para-spinal muscle area (*p* = 0.02), which reduced significantly in the those who underwent laparotomy as compared to those who did not need laparotomy. Trauma patients who underwent emergency abdominal surgery lost muscle and fat over time. **Conclusions**: Loss of muscle mass and body fat is not uncommon among young trauma patients. Patients who underwent laparotomy are more likely to be affected. Further larger studies are needed to assess the specific features in the younger trauma population and how far this can be influenced by the nutrition status and its impact on the clinical outcomes. It could be early or impending stages of sarcopenia linked to trauma patients, or just acute changes in the muscle and fat, that need further investigation and follow-up after hospital discharge.

## 1. Introduction

Trauma plays a significant role in hospital admissions, which has a substantial burden on the healthcare systems worldwide. Moreover, it remains a leading cause of death and morbidity among all age groups. It is essential to understand the response of the body and management strategies to support the recovery process after a traumatic event. The associated metabolic and neurohormonal response significantly impacts muscles and fat metabolism [1]. Malnutrition in critically ill hospitalized patients is well described, with a high overall incidence. However, the trauma patient-specific incidence is not well understood [2]. A recent systematic review reported a prevalence of malnourishment ranging between 7 and 76% in severely injured trauma patients [3]. The authors also found the baseline prevalence of malnourishment in geriatric trauma patients on admission to be in similar range (7–62%). Initially, after hospital admission due to trauma, young, injured subjects show good nutritional condition. However, they remain at high risk of malnutrition due to hyper-metabolism status (catabolic hormones), inflammatory response, hemodynamic instability, and delay of feeding due to surgical issues in the acute phase post-trauma. Therefore, malnutrition in trauma patients should be avoided, as it increases the proteolysis process, decreases immune functions, and delays wound healing [4]. Notably, the muscle loss results from bed rest and inadequate protein intake, especially in trauma patients who undergo multiple abdominal surgeries and, consequently, experience longer periods of fasting than those with isolated orthopedic trauma [5]. Accordingly, the protein deficit results in a negative nitrogen balance and muscle wasting [6]. The caloric deficits that lead to depletion of body fat and muscle mass can be considered an indication of malnutrition that could be associated with a higher rate of in-hospital complications and prolonged hospital stays [7].

Sarcopenia, as an example of mass muscle loss, is known as the age-related loss of muscle mass and function; furthermore, acute loss during the first 28 days of hospitalization after stressful events such as trauma, surgery, or sepsis is known as acute sarcopenia [8]. However, the muscle and/or fat can be reduced at any time depending on the severity of the stressor and nutrition status. It is more frequent in the elderly and critically ill/injured younger patients [9]. Earlier studies in other areas showed that muscle mass and fat changes, whether primary or secondary, impacted various outcome parameters, depending on the target population [9,10]. Most studies related to the surgical field evaluated the impact on various parameters in orthopedics [11,12,13] and cancer-related surgical out-comes [14,15,16]. Furthermore, in critically ill surgical patients, it is well described that acute illness is associated with intense muscle loss exceeding that of cancer [17,18]. Fengchan et al. reported the impact of these changes on readmission and overall survival after abdominal trauma [19]. However, it remains ambiguous whether the condition can be assessed in critically ill or ventilated patients in which the muscle strength or performance cannot be adequately evaluated. Furthermore, it is not fully explored yet the extent to which the development or degree of these changes during the hospital course can be influenced in trauma patients at earlier stages post-admission. Given how difficult it is to assess the muscles strength in polytrauma, bone fractures, painful/tender regions, open wounds, and ventilated patients, early detection of acute muscle/fat mass loss is challenging. Therefore, this study aims to explore the incidence, diagnosis, feeding, and outcome of acute muscle mass and fat changes among young trauma patients one week post-admission. 

## 2. Materials and Methods

It is a retrospective cohort study including the abdominal polytrauma patients who required hospital admission and underwent initial and follow-up abdominal computerized tomographic (CT) examination during three study periods. Patients who had incomplete laboratory data, CT scan findings, and dietary feeding information on admission and within a week were excluded. Also, patients who died on arrival or post-cardiac arrest were excluded from the analysis. Data were obtained from the Qatar National Trauma Registry (QNTR) database and electronic medical records of the patients who fulfilled study criteria and were admitted at the National Level I trauma center at Hamad General Hospital, Qatar. The Qatar National Trauma Registry is a mature database that participates in the National Trauma Data Bank and Trauma Quality Improvement Program of Committee on Trauma by the American College of Surgeons (TIQP-ACS). This trauma registry is well validated internally and externally on a regular basis.

Data collection included patients’ demographics, vital signs, body mass index (BMI), body surface area, mechanism of injury, injured region/organ, prehospital and admission Glasgow coma scale (GCS), shock index (SI), injury severity score (ISS), abbreviated injury score (AIS), laboratory and CT imaging findings (on admission and after a week), surgical interventions, intervention radiology, feeding (oral, enteral and parenteral), timing, type of feeding formula, amount of calories and protein, tolerance of feeding, ventilatory days, length of intensive care unit (ICU) and hospital stay, and in-hospital complications such as pneumonia, acute respiratory distress syndrome (ARDS), infection, acute kidney injury, thromboembolic events, and all-cause mortality. The definitions of SI (simultaneous heart rate divided by systolic blood pressure), GCS (it assesses the level of consciousness ranging between 3 to 15), AIS (it describes the anatomic injury; a severity level ranging from 1 to 6 is assigned to each traumatic injury), and ISS (it is based on an anatomic classification of injury severity of the AIS but combines the severity levels in a single value ranging from 0 to 75; the higher the score, the higher the injury severity), were described previously [20,21,22]. Abdominal polytrauma was defined as an abdominal injury in which abdomen AIS ≥ 1 plus ISS more than 9.

The diagnosis of sarcopenia is based on the combination of functional parameters and skeletal muscle mass assessment and is graded for severity with performance measurements [23]. However, we did not rely on this definition perfectly, as it was not age-related in our study, we did not assess functionality early in polytrauma or ventilated patients, and we repeated the CT after one week only (not 4 weeks). Therefore, we aimed to assess the early, or acute, muscle and fat changes within a week. The alternate modality to assess the muscle mass includes imaging techniques such as dual-energy X-ray absorptiometry (DXA) and bioelectrical impedance analysis (BIA). In critically injured patients, the clinical tools commonly used to assess muscle function, such as the handgrip dynamometer and the chair stand test, are not applicable. The abdominal CT scan is the gold standard imaging modality in trauma settings. In this study, mass of muscle and fat changes was evaluated through the assessment of abdominal CT scan imaging to detect fat depletion in trauma patients, using muscle mass/fat area assessment as a surrogate.

Patients were scanned on Siemens Somatom 64 slice CT scanner. The axial section of the abdomen at the lower endplate of the L3 vertebra was selected. The image was processed on Fuji Synapse 3D software for fat and muscle analysis. The cross-sectional areas of subcutaneous fat, visceral fat, and total fat were calculated using automatic processing by the fat analysis software with manual fine-tuning. The software also automatically calculated the cross-sectional area of the psoas muscles. The total abdominal and para-spinal muscle cross-sectional areas were calculated by tracing the software’s strengths manually. Computed tomography abbreviated assessment of sarcopenia for trauma (CAAST) was obtained by measuring the psoas muscle cross-sectional area adjusted for height and weight [24]. In trauma settings, the muscle quantity was considered a reliably measurable parameter as a surrogate for the presence of sarcopenia. 

This study was granted approval from the Research Ethics Committee of the Medical Research Center, Hamad Medical Corporation (IRB # MRC-01-17-130). A waiver of in-formed consent was granted, as there was no direct contact with patients and data were anonymously collected.

**Statistical analysis:** Data were presented as appropriate proportions, medians, or mean ± standard deviation (SD). Dietary order details were compared on admission and one week following the injury. Also, the laboratory and radiological findings were compared at the presentation and after one week. The percentage of change in the total fat area, visceral fat area, and subcutaneous fat area, as well as the total abdominal skeletal muscle area, total psoas muscle area, and para-spinal muscle calculated from CT scan findings, were correlated with the first dietary order on admission (oral versus enteral), and compared based on the GCS (GCS ≤ 8 vs. GCS > 8), systolic blood pressure (SBP < 90 versus SBP ≥ 90), ISS (ISS ≤ 15 vs. ISS > 15), and laparotomy (yes vs. no). Furthermore, the changes in the laboratory findings over time were correlated with the first dietary order on admission (oral versus enteral) and Glasgow coma score (GCS ≤ 8 vs. GCS > 8). Differences in the continuous variables were analyzed using Student’s *t*-test. For skewed continuous data, a non-parametric Mann–Whitney test was performed. Chi-square test was used for categorical variables comparison. Two-tailed *p* values of <0.05 were considered significant. Data analysis was carried out using the Statistical Package for Social Sciences version 21.0 (SPSS Inc. USA). 

## 3. Results

A total of 138 abdominal polytrauma patients fulfilled the inclusion criteria. Patients were predominantly younger (32.8 ± 13.5 years), the majority were males (92%), and all had mean BMIs of 26.8 ± 5.5 (Table 1). The most frequently injured anatomical regions were the abdomen, followed by the chest, spine, pelvis, extremities, and head. The mean ISS was 24.6 ± 11.4. One-third of the patients had solid organ injuries, with the majority having isolated liver (40.9%) and splenic (29.5%) injuries, while multiple SOI was observed in 18% of the cases. Operative interventions were performed on two-thirds of the patients, with abdominal surgery being the most common procedure (43%), followed by orthopedic surgeries (34%) and neurosurgical procedures (8.1%). Intervention radiology was used in 23% cases.

One-third of the patients sustained bowel injuries necessitating resections (small bowel in 47.5% of the cases, large bowel in 30%, and both small and large in 20% of the cases). Ten patients underwent damage control resuscitation; the overall rate of anastomotic leak was 2.5%. The most frequent in-hospital complication was pneumonia, followed by sepsis, wound infection, and acute respiratory distress syndrome (ARDS). The median hospital length of stay was 31 days, the ICU was 16 days, and ventilator days was 12. The overall mortality rate was 5.8%.

Table 2 shows the first dietary order details on admission and after the first week. On admission, 56% received oral feeding, and this slightly increased to 58.4% after the first week. Enteral feeds were used for the remaining, except for two patients: one needed parenteral nutrition on admission and another needed it after the first week. Nasogastric tubes were the most common enteral access on admission and after the first week. The enteral formula was the standard 1:1 (1 mL provides 1 kilocalorie) in the majority (95%) of cases. On admission, the elemental formula was used in 67%, dropping to 17% after the first week, where the standard formula was more frequent (39%). High-protein formulas were only used after the first week in 20%. On admission, the median infusion rate was low— 20 mL/h versus 70 mL/h after one week—which provided fewer calories initially (median 600 K) and increased after one week (2000 K), and the median protein amounts were 51 g initially, which increased to 113 g after the first week. It was well tolerated, and the nutritional target was sufficient on admission in 40%, which increased to 71% after one week. These findings reflected a clear undernutrition achievement on admission and after the first week. Regardless of the increased needs due to trauma, there were multiple other factors. The associated body responses, complications, and interruption of feeding were related to nil per os (NPO) orders for procedures or feeding intolerance, particularly in those who underwent abdominal surgeries with postoperative ileus or NPO standard orders if bowel resection and anastomosis were used. Figure 1 shows the axial CT image after processing, demonstrating subcutaneous fat, visceral fat, psoas muscle, and para-spinal muscles.

Table 3 compares the laboratory and CT findings after hospital admission and one-week post injury and shows drops in the average levels of blood hemoglobin, creatinine, albumin, troponin, myoglobin, and lactate. Also, there was a decline in the CT scan findings of fat and muscle mass areas.

Table 4 shows a comparison between oral and enteral feeds upon admission and after one week, using the CT measures (fat and muscles). The percentage change in the total fat area, visceral fat area, subcutaneous fat area, total abdominal skeletal muscle area, and para-spinal muscle area were comparable among the two groups. However, the percentage change in the total psoas muscle area was significantly reduced after one week of admission in patients on enteral feed as compared to those in the oral feeding group. Notably, the oral feeding usually reflected less severely injured patients. 

Table 5 compares the CT scan findings for fat and muscle areas based on the following: (1) severe TBI (GCS ≤ 8) vs. mild-moderate TBI (GCS > 8), (2) on-admission systolic blood pressure (hypotensive vs. non-hypotensive patients), (3) injury severity score (ISS ≤ 15 vs. ISS > 15), and (4) the need for laparotomy (yes vs. no). Interestingly, all the subgroup analysis showed no statistically significant differences in the percentages of changes in the CT scan findings except for the total psoas muscle area (*p* = 0.001) and para-spinal muscle area (*p* = 0.02), which reduced significantly in those who underwent laparotomy as compared to those who did not need laparotomy. 

Table 6 compares the laboratory parameters based on the dietary admission order (whether oral or enteral) and the Glasgow coma score (GCS ≤ 8 vs. GCS > 8). The only observed statistically significant difference between orally fed and enterally fed upon admission was that the platelet count and the international normalized ratio (INR) measures were lower in the orally fed. On the other hand, the comparison based on GCS did not show significant difference among the two groups except for two parameters. The percentage change in the INR was higher in the severe group (GCS ≤ 8; *p* = 0.002), while the prothrombin time was lower in severely injured (GCS ≤ 8; *p* = 0.003) patients.

## 4. Discussion

The current study is the first of its kind in the Middle Eastern region, providing valuable insights into the incidence, diagnosis, feeding, and outcome of acute muscle and fat loss among young trauma patients based on serial CT scan, laboratory findings, and feeding status. Although it does not fulfil the typical criteria of non-trauma sarcopenia, it addresses a similar condition in a short time, encouraging further exploration. It is important to highlight that trauma patients are recognized for experiencing intricate processes of repair and recovery. In this study, muscle and fat loss were based on CT assessments that measured fat and muscle loss during the hospital course. The possible justification to rely on this is the post-trauma hyper-metabolic surge, which is displayed as an increased energy demand and, simultaneously, increased breakdown/loss. There is a high incidence of under-nutrition during the early phase of trauma; also, disuse atrophy, ongoing inflammation processes, and potentially arising post-traumatic complications are other plausible explanations for these complex trauma-related body changes [25,26]. We reported interval changes in the CT scan-driven indices of muscle and fat and the nutritional status details. The results reflected a loss of muscle mass and fat depletion based on the admission GCS, hypotension, ISS, and laparotomy.

Most of the literature related to sarcopenia in trauma investigated older people who underwent orthopedic surgery or major abdominal surgery The definition of sarcopenia entails measures of three parameters (muscle strength, muscle quantity/quality, and physical performance with the three-denoting severity) [27]. Despite limitations in its diagnosis in young-age people, sarcopenia gained significant recognition in research in critically ill patients and became a well-accepted diagnosis. 

The etiology of muscle wasting in critically ill younger patients is complex and influenced by multiple factors. The development of muscle loss is linked to factors such as reduced physical activity, the presence of proinflammatory cytokines, and diminished protein synthesis [28,29]. The major concern is the accelerated decline in muscle mass and strength, which can affect younger patients who have suffered severe injuries that could potentially impact the clinical outcome to the same well-known effects of sarcopenia in the elderly. 

Patients with abdominal trauma often suffer from interrupted or delayed enteral feeding due to the risk of feeding intolerance and feeding-associated complications [30]. Also, abdominal trauma patients may have NPO orders for observation or repetitive surgery. Moreover, if young patients undergo a laparotomy, they often continue fasting until the bowel function returns to normal. However, this period of fasting can be prolonged if complications such as leaks, ileus, or surgical site infections occur, categorizing them as a high-risk group within the trauma population [31]. 

A well-investigated subgroup of trauma is burn patients, in which adequate nutrition is known to have a positive impact on clinical outcomes despite significant metabolic stress. A review on metabolism and nutrition in critically ill burn patients showed that early feeding strategies, optimization, and personalization of nutritional goals improve clinical outcomes [32]. The acute phase response is a common physiological reaction observed in cases of trauma, infections, and burns [33]. Consequently, the effects of energy expenditure and catabolism can be increased by up to 50%, depending on the severity of the injury. This hypermetabolic state is essential for the body’s recovery from such incidents, but, if disturbed, it can lead to morbidity and mortality [1,34]. The hypermetabolic state results in decreased synthesis and depletion of the glycogen reserves and muscle protein within the body. Additionally, there is a surge in lipid metabolism, particularly between days 3 and 7 post injury. It is well known that the subsequent phase of hypermetabolism is characterized by insulin resistance and hyperglycemia [35].

In our study, the majority of the patients (43%) underwent abdominal surgery, and most of them had resection of the small bowel (47.5%), followed by large bowel resection (30%) and combined large and small bowel resection (20%). Notably, 2.9% developed complications such as leaks. 

Interestingly, laparotomy patients showed a significant loss in the total psoas and para-spinal muscle areas in our study cohort. Muscle bulk and fat loss are related to trauma-associated hypermetabolism. The observed loss in muscle mass among laparotomy patients indicates a depletion of physiological reserve in such patients due to inadequate nutritional status [36]. 

Surprisingly, a subgroup of fasting patients showed a significant loss in psoas muscle mass instead of fat. This could be attributed to the orthopedic surgery, neurosurgical interventions, planned extubation, interventional procedures, massive reflux, and imaging procedures such as CT scan or MRI. These findings may be influenced by age, as most studies discussed orthopedic trauma in elderly patients. Unlike these studies, our cohort constitutes younger patients with an average age of 33 years. 

In our study, the mean total daily calorie intake was 600 kcal. The enteral feeding was 92.6% well-tolerated, as indicated by the lack of vomiting or aspiration of large amounts of fluid from the stomach within 8 h, a standardized measure in the trauma ICU. However, in the first week after hospital admission, enteral feeding in trauma patients was insufficient in almost half of the individuals. During the second week, improvement concerning the feeding was observed, which can be attributed to the reduced necessity of pausing the enteral feeding due to less frequent imaging procedures, laparotomies, or other investigations. This finding agrees with earlier studies on the fact that only a few achieved timely nutritional supports and, thus, malnutrition continued in most investigations [37,38]. However, malnutrition does not develop overnight, as some of the patients may be malnourished from the beginning, so the current practices do not provide adequate requirements. Further examination of the coagulation parameters during the hospital stay revealed the presence of trauma-induced coagulopathy, a well-known and previously described condition [39]. 

A recent article demonstrated a clear approach to avoid or reduce muscle and fat loss [9]. The authors suggested that if enteral nutrition needs to be stopped or is not possible for other reasons, it should be continuously and efficiently replaced with parenteral nutrition [9]. Measurement and adequate replacement of vitamins (such as vitamin D) and calcium, as well as exercise and, if possible, electrophysiological stimulation, should be consequently applied [9,40].

**Limitations:** This study has some limitations due to its retrospective design, which has potential for missing data and selection bias. Also, the inability to have long-term muscle function and performance assessment. We believe that routine checks on muscle mass/function and performance are necessary to document the true prevalence of this condition. As addressed earlier, a regular assessment of muscle strength should be started as soon as a patient is awake and cooperative. To mitigate and minimize the potential impacts of muscle loss on the clinical outcomes of high-risk patients, it is crucial to take proactive measures. This includes implementing a combination of well-designed physical exercises and nutritional supplementation. This unexplored territory would greatly benefit from collaboration among international trauma centers. The gender difference was not part of the analysis due of the paucity of female subjects. According to our trauma database, the male to female ratio in overall trauma, or particularly in blunt abdominal trauma, is 9 to 1 and [21,22]. We could not assess these changes after 28 days by CT scan and functionality (to fulfill the criteria of sarcopenia) as there was no clinical indication to expose these patients to radiation and as most of the cases are discharged before the 28 days. Moreover, as it was a retrospective study and no protocol was established, this was not possible to attain in the trauma patients. Therefore, we may consider it as an early or impending stage of sarcopenia linked to trauma patients or just acute changes in the muscle and fat that need further investigation and follow-up after hospital discharge.

Although there are guidelines to address nutritional support for the critically injured [41], no substantial evidence links increasing delivery to better outcomes [42]. The evidence is unclear on whether interventions for matching the requirements or the standard underfeeding are better [43]. The pilot trial known as TOP-UP discovered that a specific subset of patients, those with a BMI below 25 or above 35, would derive benefits from parenteral supplemental nutrition [44]. However, this cohort does not provide answers to questions concerning enteral versus parenteral nutrition or the optimal timing (early versus late) for implementation in trauma patients. Therefore, to overcome the limitations of this observational study, there is a need to capture more details about the muscle strength and function, nutrition, underlying hormone response, nitrogen balance assessment, exact times, and reasons for nutrition pausing. 

Furthermore, trauma staff awareness of the routine use of validated nutritional assessment tools to identify and quantify the malnutrition problem should be raised. Finally, the definition and early recognition of sarcopenia remain not well-settled and underutilized in the plans and management of trauma patients, especially among the young population [45,46]. A recent systematic review of 18 studies recommended incorporating radiographic assessment of sarcopenia into patient management plans in addition to using a consistent definition of sarcopenia to strengthen and optimize its applicability and efficiency in clinical trauma care [46]. In our study, we opted to review the CT scan findings on admission and after one week to have a better definition and understanding of the muscle and fat changes as early as possible; however, we might underestimate the real proportion of patients who may develop typical sarcopenia later than the first week post-admission. Therefore, serial assessment of muscle strength and function might overcome this limitation as well as the lack of serial CT scans.

## 5. Conclusions

This analysis suggests that early loss of muscle mass and body fat are not uncommon among young trauma patients. The findings also reflect the undernutrition status on admission and after the first week. Patients who underwent laparotomy were more likely to experience a significant reduction in total psoas muscle area and para-spinal muscle area, possibly due to undernutrition. Therefore, implementing quality assurance projects that ensure the timely and adequate provision of nutritional support, effective monitoring, and proper documentation is vital for optimizing the recovery of critically injured trauma patients. These initiatives also play a crucial role in identifying appropriate interventions that adapt promptly to the significant trauma-induced responses in muscle and fat metabolism. Moreover, the definition and early management of this clinical entity needs to be clear, and awareness needs to be raised in trauma patients. Further larger studies are needed to assess whether these findings can warn of or indicate pending sarcopenia in the younger trauma population and how far this can be influenced by the nutrition status and its impact on the patient’s clinical outcomes.

## Figures and Tables

**Figure 1 diseases-11-00120-f001:**
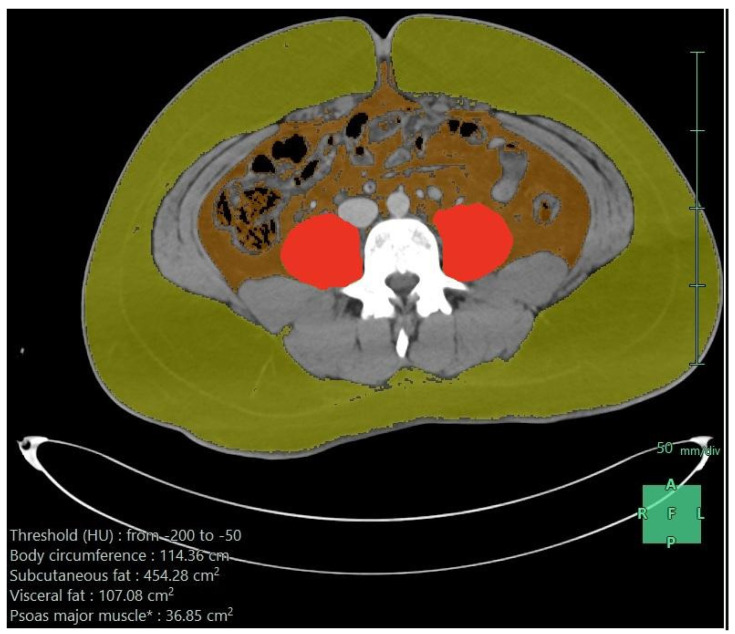
Axial CT scan image after processing (subcutaneous fat in greenish yellow, visceral fat in brown color, and psoas muscle in red color); the measurements are given in the lower part of the image.

**Table 1 diseases-11-00120-t001:** Demographics, clinical presentation, management, complications, and outcome of trauma patients (n = 138).

Variables	Value	Variables	Value
Age (mean ± SD)	32.8 ± 13.5	**Type of surgery**	
Gender		Abdominal	43 (43.4%)
**Male**	127 (92.0%)	Orthopedic	34 (34.3%)
**Females**	11 (8.0%)	Neurosurgery	8 (8.1%)
Body mass index	26.8 ± 5.5	Others	14 (14.1%)
Body surface area	1.86 ± 0.24	**Bowel resection**	40 (29.0%)
Shock index at the scene	0.83 ± 0.26	Resection alone	10 (25.0%)
GCS at Scene	15 (3–15)	Resection and anastomosis	30 (75.0%)
Shock index on admission	1.02 ± 0.46	**Location of bowel resection**	
GCS on admission	15 (3–15)	Small bowel	19 (47.5%)
Injury severity score	24.6 ± 11.4	Large bowel	12 (30.0%)
Abdomen AIS (n = 114)	3.1 ± 1.0	Both bowels	8 (20.0%)
Chest AIS (n = 88)	2.9 ± 0.7	Stomach	1 (2.5%)
Spine AIS (n = 58)	2.3 ± 0.9	Bowel leak	4 (2.9%)
Pelvis AIS (n = 52)	2.3 ± 0.7	**Complications**	
Upper extremity AIS (n = 50)	2.0 ± 0.2	Pneumonia	37 (26.8%)
Lower extremity AIS (n = 43)	2.6 ± 0.5	Sepsis	28 (20.3%)
Head AIS (n = 35)	3.9 ± 0.9	Wound infection	23 (16.7%)
Spinal cord injury	6 (4.3%)	ARDS	18 (13.0%)
Solid organ injury	44 (31.9%)	Urinary tract infection	12 (8.7%)
Liver	18 (40.9%)	Acute kidney injury	11 (8.0%)
Spleen	13 (29.5%)	Deep vein thrombosis	3 (2.2%)
Multiple	8 (18.2%)	Pulmonary Embolism	2 (1.4%)
Kidney	3 (6.8%)	**Hospital length of stay**	31.5 (8–166)
Pancreatic injury	2 (4.5%)	**ICU length of stay**	16 (1–163)
Intervention radiology	32 (23.2%)	**Ventilatory days**	12 (2–54)
Surgical intervention	99 (71.7%)	**Mortality**	8 (5.8%)

**Table 2 diseases-11-00120-t002:** First dietary order details on admission and after the first week.

	On Admission	after the First Week	*p* Value
**Oral ***	77 (55.8%)	80 (58.4%)	0.61 for all
**Enteral**	60 (43.5%)	55 (40.1%)	
- Nasogastric tube	37 (60.7%)	40 (72.7%)	0.001 for all
- Nasojejunal tube	9 (14.8%)	15 (27.3%)	
- Orogastric tube	14 (23.0%)	0 (0.0%)	
**Parenteral**	1 (0.7%)	2 (1.5%)	1.00
**Diet Order for oral intake**	1 (1–7)	-	-
**Strength Of formula for enteral feed ^^^**			
1 mL:1 kcal	57 (95.0%)	-	-
1 mL:1.8 kcal	3 (5.0%)	-	-
**Infusion rate (mL/h)**	20 (10–80)	70 (10–100)	0.001
**Type of formula used**			0.001 for all
Standard	17 (28.3%) **	21 (38.9%) ***	
Elemental	40 (66.7%)	9 (16.7%)	
Renal	3 (5.0%)	3 (5.6%)	
High protein	-	11 (20.4%)	
Pulmonary	-	2 (3.7%)	
Wound healing	-	3 (5.6%)	
Diabetic	-	4 (7.4%)	
Hepatic formula	-	1 (1.9%)	
**Amount of calorie (kg/BW)**	600 (240–2590)	2000 (240–4000)	0.001
**Amount of protein (gm) (kg/BW)**	51 (8–143)	113 (8–231)	0.001
**Tolerance of feed ^†^**	126 (92.6%)	130 (94.9%)	0.44
**Not Sufficient**	76 (55.9%)	39 (28.7%)	0.001 for all
**Sufficient**	60 (44.1%)	97 (71.3%)	

* median number of NPO orders during admission [8 (1–41)]; ^^^ ** available for 60 cases; *** available for 54 cases; ^†^ available for 136 cases.

**Table 3 diseases-11-00120-t003:** Laboratory and radiological findings on admission and after one week of the injury.

	on Admission	after One Week	*p* Value
**Laboratory findings**			
WBC count	17.3 ± 7.5	12.8 ± 5.1	0.001
Hemoglobin level	12.5 ± 3.4	9.7 ± 1.5	0.001
Platelet count	256 ± 85.4	275 ± 131	0.11
Serum creatinine	92.6 ± 29.6	69.7 ± 60.1	0.001
Total protein	52.6 ± 10.4	53.3 ± 10.3	0.05
Serum albumin	31.3 ± 6.2	25.2 ± 5.8	0.001
International normalized ratio	1.13 ± 0.16	1.07 ± 0.83	0.001
Partial thromboplastin time	27.0 ± 7.7	28.2 ± 4.8	0.004
Serum lactate	3.7 ± 2.6	1.48 ± 0.89	0.001
C-Reactive protein	159.5 ± 125.2	-	-
Serum myoglobin	855 (8–4882)	124 (22–2284)	0.001
Serum troponin	62.5 (2–1443)	29.5 (4–629)	0.03
**CT scan findings**			
Total fat area	239.3 ± 161.4	213.5 ± 150.6	0.001
Visceral fat area	94.8 ± 74.3	81.6 ± 64.3	0.001
Subcutaneous fat area	144.5 ± 109.2	131.9 ± 104.3	0.001
Total abdominal skeletal muscle area	156.9 ± 39.4	149.9 ± 37.5	0.001
Total psoas muscle area	27.1 ± 7.7	25.0 ± 7.3	0.001
Para-spinal muscle area	64.2 ± 13.7	59.6 ± 13.4	0.001

**Table 4 diseases-11-00120-t004:** Computed tomography scan findings based on dietary order.

	Oral (n = 76)			Enteral (n = 60)			*p*-Value *
	on Admission	after One Week	Percentage Change	on Admission	after One Week	Percentage Change	
Total fat area	222.4 (41–700.8)	214.0 (17.8–585.5)	−5.65 (−58–0.4)	217.6 (10.2–891.0)	180.2 (7.4–772.0)	−7.2 (−90.8–0.8)	0.47
Visceral fat area	78.3 (8.1–263.8)	67.3 (6.8–253.1)	−6.9 (−63.6–27.1)	84.3 (4.6–382.2)	74.0 (3.4–294.3)	−8.3 (−92.2–8.8)	0.62
Subcutaneous fat area	130.3 (26.8–539.7)	123.4 (10.2–484)	−5.1 (−63–2.6)	110.1 (5.6–586.0)	104.5 (4.0–549.0)	−5.3 (−88.7–15.9)	0.99
Total abdominal skeletal muscle area	150.4 (14.2–222.5)	140.9 (35–208.9)	−4.2 (−28.8–739)	158.2 (94.5–356.4)	146.3 (82.3–354.0)	−2.2 (−38.6–28.0)	0.51
Total psoas muscle area	27.9 (9.2–46.8)	24.0 (12.4–54.3)	−5.9 (−31.3–17.0)	26.1 (12.1–54.0)	23.6 (12.3–44.9)	−8.9 (−98.3–13.0)	0.02
Para-spinal muscle area	63.3 (32.8–89.1)	58.1 (28.2–89.9)	−6.3 (−31.3–17.0)	63.8 (39.9–134.7)	59.0 (29.1–129.7)	−4.7 (−32.5–11.5)	0.82

* *p*-value is for percentage change between the two groups (oral vs. enteral).

**Table 5 diseases-11-00120-t005:** Change in the CT scan findings in different settings.

	GCS ≤ 8 (n = 32)	GCS > 8 (n = 102)	*p*-Value
Total fat area	−5.8 (−50.2–1.1)	−6.2 (−90.8–0.4)	0.63
Visceral fat area	−8.7 (−56.7–8.8)	−6.8 (−92.2–27.1)	0.70
Subcutaneous fat area	−5.4 (−48.7–4.3)	−5.3 (−88.7–15.9)	0.52
Total abdominal skeletal muscle area	−2.9 (−26.2–7.7)	−3.6 (−38.6–739.4)	0.87
Total psoas muscle area	−11.0 (−59.5–4.1)	−6.2 (−98.3–107.6)	0.05
Para-spinal muscle area	−5.4 (−23.3–11.1)	−5.8 (−32.5–17.0)	0.74
	**SBP < 90 (n = 13)**	**SBP ≥ 90 (n = 118)**	
Total fat area	−5.3 (−26.0–1.6)	−6.2 (−90.8–0.4)	0.78
Visceral fat area	−4.9 (−50.6–5.2)	−7.2 (−92.2–27.1)	0.36
Subcutaneous fat area	−6.0 (−29.4–2.4)	−4.9 (−88.7–15.9)	0.40
Total abdominal skeletal muscle area	−1.3 (−8.0–5.9)	−3.5 (−38.6–739.4)	0.05
Total psoas muscle area	−5.8 (−24.9–13.5)	−6.8 (−98.3–107.6)	0.56
Para-spinal muscle area	−3.1 (−10.5–11.1)	−5.5 (−32.5–17.0)	0.90
	**ISS ≤ 15 (n = 29)**	**ISS > 15 (n = 109)**	
Total fat area	−7.9 (−57.5–0.1)	−6.2 (−90.8–0.4)	0.43
Visceral fat area	−8.6 (−63.6–1.0)	−6.9 (−92.2–27.1)	0.24
Subcutaneous fat area	−5.8 (−63.0–0.8)	−5.1 (−88.7–15.9)	0.37
Total abdominal skeletal muscle area	−4.1 (−14.7–3.3)	−2.9 (−38.6–739.4)	0.81
Total psoas muscle area	−5.6(−35.3–107.6)	−7.5 (−98.3–55.9)	0.18
Para-spinal muscle area	−4.9 (−20.6–17.0)	−5.8 (−32.5–11.5)	0.19
	**No-Laparotomy (n = 89)**	**Laparotomy (n = 49)**	
Total fat area	−5.2 (−54.9–0.4)	−8.9 (−90.8–0.9)	0.06
Visceral fat area	−7.1 (−90.3–27.1)	−7.0 (−92.2–2.5)	0.40
Subcutaneous fat area	−4.8 (−48.7–15.9)	−6.0 (−88.7–3.6)	0.11
Total abdominal skeletal muscle area	−2.9 (−28.8–739.4)	−3.5 (−38.6–28.0)	0.51
Total psoas muscle area	−5.5 (−59.5–107.6)	−13.0 (−98.3–13.0)	0.001
Para-spinal muscle area	−4.6 (−31.3–17.0)	−7.2 (−32.5–11.5)	0.02

**Table 6 diseases-11-00120-t006:** Change in the laboratory values based on the dietary order in trauma patients.

Variable	Oral (n = 76)	Enteral (n = 60)	*p* Value
WBC count	−30.4 (−100–780)	−36.3 (−100–908)	0.58
Hemoglobin	−22.6 (−100–61.8)	−26.8 (−100–61.8)	0.50
Platelet count	15.3 (−100–319)	−16.0 (−100–324)	0.006
Serum creatinine	−32.9 (−100–40)	−33.7 (−100–316.3)	0.83
Total protein	0.80 (−100–46)	1.0 (−100–96)	0.20
Serum albumin	−18.9 (−100–105.6)	−25.8 (−100–70.6)	0.19
International normalized ratio	−9.1 (−100–900)	−16.7 (−100–11)	0.01
Partial thromboplastin time	12.7 (−100–64)	−4.8 (−100–104)	0.03
Serum lactate	−76.8 (−100–153)	−84.9 (−100–143)	0.71
Serum myoglobin	−100 (−100–823)	−100 (−100–35.9)	0.47
Serum troponin	−100 (−100–613)	−100 (−100–476)	0.70
	**GCS ≤ 8 (n = 32)**	**GCS > 8 (n = 102)**	
WBC count	−35.9 (−100–908)	−33.7 (−100–780)	0.75
Hemoglobin value	−20.4 (−100–61.8)	−23.8 (−100–31.9)	0.18
Platelet count	−13.8 (−100–324)	10.4 (−100–319)	0.15
Serum creatinine	−33.7 (−100–101)	−33.2 (−100–316)	0.97
Total protein	6.3 (−100–96)	0.0 (−100–59)	0.12
Serum albumin	−21.3 (−100–106)	−21.1 (−100–105)	0.90
International normalized ratio	−18.2 (−43–10)	−9.1 (−100–900)	0.002
Partial thromboplastin time	−10.1 (−83–53)	13.8 (−100–104)	0.003
Serum lactate	−90.3 (−100–143)	−77.8 (−100–153)	0.54
Serum myoglobin	−100 (−100–33.5)	−100 (−100–823)	0.55
Serum troponin	−100 (−100–317)	−100 (−100–613)	0.98

## Data Availability

Data were given in the manuscript, tables, and figures.

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
