# Peer review of "Acute Changes in Body Muscle Mass and Fat Depletion in Hospitalized Young Trauma Patients: A Descriptive Retrospective Study"

_diseases, 2023, doi:10.3390/diseases11030120_

Round 1

Reviewer 1 Report

Dear Authors,

My comments on the current manuscript indicate and refer to the following points.

All changes should be in red font color and highlighted with yellow. They should also be shown in the text.

Abstract Section:

In general, the current abstract effectively outlines the objectives, methods, findings, and conclusions of the study on acute changes in body muscle mass and fat depletion in hospitalized young trauma patients. It highlights the significance of the topic, provides relevant background information, and offers insights into the association between muscle mass loss, nutritional status, and patient outcomes. The abstract concludes by calling for further research to better understand these acute changes and their impact on clinical outcomes. However, to enhance the abstract, it could benefit from providing more specific details about the actual findings from the muscle and fat analysis to give readers a better understanding of the results obtained. Adding these details would improve the overall comprehensiveness and clarity of the abstract.

Introduction Section:

The introduction effectively highlights the significance of trauma as a major contributor to hospital admissions and its impact on healthcare systems worldwide. It also emphasizes the importance of understanding the body's response and management strategies following traumatic events. With a few additional points of clarification and some specific examples, the introduction can be further enhanced.

1) The mention of malnutrition in critically ill hospitalized patients and the need for understanding its incidence in trauma patients is a relevant point. However, it would be beneficial to briefly explain why trauma patients may be at higher risk for malnutrition compared to other critically ill populations. This would provide further context and help readers understand the importance of studying this specific group.

2) Lines 58-64: The range of prevalence of malnourishment mentioned in severely injured trauma patients and geriatric trauma patients adds valuable information. To strengthen this point, consider briefly discussing the potential consequences of malnutrition in trauma patients, such as increased complications and prolonged hospital stays.

3) Lines 65-67: The mention of sarcopenia and acute sarcopenia as examples of muscle loss is relevant and helps to explain the concept in the context of trauma patients. Consider adding a brief definition or explanation to ensure readers have a clear understanding of these terms.

4) Lines 77-81: The mention of the need to explore the assessment and extent of muscle mass and fat changes in critically ill or ventilated patients is interesting. Consider providing a brief explanation or rationale for the challenge in assessing muscle strength or performance in these patients to further clarify the research gap being addressed.

5) Line 60: Acaccordingly change to "Accordingly"

Materials and Methods Section:

Overall, the Materials and Methods section provides a detailed and well-described methodology for the study. It includes relevant inclusion/exclusion criteria, data sources, variables collected, imaging techniques, and ethical considerations. Consider reviewing the section for any minor grammar or formatting improvements, but the content itself appears robust.

1) The reference to previously described definitions for shock index (SI), Glasgow Coma Scale (GCS), Injury Severity Score (ISS), and Abbreviated 100 Injury Score (AIS) is useful for understanding these measurement tools. However, it would be beneficial to provide a concise overview or definition of each measure within the Materials and Methods section itself for readers' convenience.

2) The clear definition of abdominal polytrauma as an abdominal injury with an abdomen AIS ≥1 and ISS greater than 9 is appreciated. Providing definitions for key terms within the study's context helps readers understand the inclusion criteria.

Results Section:

Although your results are well described in accordance with your study, you have a large number of tables. However, I recommend merging some of them.

Discussion Section:

1) Lines 269-270: A few articles discussed the topic … than sarcopenia. Please use reliable references.

2) Lines 275-277: The prevailing belief is that ... insufficient physical activity. Please check your sentence and grammar.

3) Lines 280-284: Please use reliable reference(S).

4) Limitation: Overall, the limitations are well-addressed and provide valuable insights for improving future research in the field.

Conclusion Section:

This section provides a good summary of the study's findings, emphasizes the importance of nutritional support and monitoring, and highlights the need for further research.

Please verify the English grammar. 

Author Response

I would like to thank you for the great effort to improve this manuscript. All the comments have been addressed and attached here.

Reviewer 2 Report

The manuscript descibes a retrospective analysis of muscle and fat loss assessed by CT imaging of the abdomen at L3, in major trauma victims. The data were assessed upon admission and after 1 week, the loss of muscle and fat is thus very acute. Mnay other clicnial and biomedcial parameters are also considered as well as nutritional provision. I have the following suggestions for improvement:

1. The data in the main mansucriupt are easy to follow but are not well presented in the abstract. For exmaple the focus of the manuscript is on muscle mass and fat content yet these are not descrebd in any detail in the abstract. I would reocmmend a major revision of the abstract to include data such as % msucle and fat loss and associated p values, that psoas msucle seemed most affected by different nutritional modes. The comment re platelet count and INR could be removed as less important to the focus of the manuscript.

2. I am not sure figure 1 is necessary.

3. Line 187-188 states that undernutrition was present  on admisison and week 1, but by week 1 71% of patients received optimal nutrition. Perhaps this should be changed to state just upon admission.

4. In the concluson of the abstract and in the main document the authors stet "Loss of muscle mass and body fat is not uncommon among young trauma patients and has worse outcomes". yet as far as I cna see the study doe snot look at patietn outcomes, these sttaement must be remoived.

5. The discussion is very lengthy and would benefit from shortening. For example the disucssion on elderly sarcopenia is not necessary and discussion around inability to measure muscle function can also be shortened, as this is rather obvious.

Qualty of english is acceptbale, the main point is that in several places the authors split words, such as admission to ad-mission, this must be corrected throughout.

Author Response

Thank you for these important comments, i addressed all in the attached file

Reviewer 3 Report

Good paper and well written.

Needs a thorough reread and edit to eliminate sentence construction problems, syntax errors, spelling mistakes and grammar errors.

All Tables should have the p-values that reach significance emboldened.

Good paper - minor errors only in sentence construction, grammar and use of syntax. These should be corrected. Needs a good reread and edit.

Author Response

Thank you for this positive feedback , we have addressed your comment in the revised manuscript